# Dir-MUSIC Algorithm for DOA Estimation of Partial Discharge Based on Signal Strength Represented by Antenna Gain Array Manifold

**DOI:** 10.3390/s22145406

**Published:** 2022-07-20

**Authors:** Wencong Xu, Bingshu Chen, Yandong Li, Yue Hu, Jianxun Li, Zijing Zeng

**Affiliations:** 1Department of Automation, Shanghai Jiao Tong University, No. 800 Road Dongchuan, Shanghai 200240, China; Xuwc@sjtu.edu.cn (W.X.); jx_li@hotmail.com (J.L.); 2Department of Electrical Engineering, Shanghai Jiao Tong University, No. 800 Road Dongchuan, Shanghai 200240, China; bingshuchen@sjtu.edu.cn (B.C.); l215121659@sjtu.edu.cn (Y.L.); zijingzeng@sjtu.edu.cn (Z.Z.)

**Keywords:** partial discharge(PD), directional multiple signal classification (Dir-MUSIC), strength intensity information, spiral circular array

## Abstract

Inspection robots are widely used in the field of smart grid monitoring in substations, and partial discharge (PD) is an important sign of the insulation state of equipment. PD direction of arrival (DOA) algorithms using conventional beam forming and time difference of arrival (TDOA) require large-scale antenna arrays and high computational complexity, making them difficult to implement on inspection robots. To address this problem, a novel directional multiple signal classification (Dir-MUSIC) algorithm for PD direction finding based on signal strength is proposed, and a miniaturized directional spiral antenna circular array is designed in this paper. First, the Dir-MUSIC algorithm is derived based on the array manifold characteristics. This method uses strength intensity information rather than the TDOA information, which could reduce the computational difficulty and the requirement of array size. Second, the effects of signal-to-noise ratio (SNR) and array manifold error on the performance of the algorithm are discussed through simulations in detail. Then, according to the positioning requirements, the antenna array and its arrangement are developed and optimized. Simulation results suggested that the algorithm has reliable direction-finding performance in the form of six elements. Finally, the effectiveness of the algorithm is tested by using the designed spiral circular array in real scenarios. The experimental results show that the PD direction-finding error is $3.39^{{}^{\circ}}$, which meets the need for partial discharge DOA estimation using inspection robots in substations.

## 1. Introduction

Partial discharge [1] is one of the main causes of power equipment insulation failure. An ultra high frequency (UHF) detection method [2] using the UHF component of the PD signal is widely used due to its strong anti-interference ability and high sensitivity. In recent years, many UHF detection methods based on time difference of arrival (TDOA) have been proposed to detect and locate the PD source using an omni-directional UHF sensor array in substations, which obtain fruitful results [3,4,5,6,7,8,9,10]. These methods mainly rely on the TDOA of UHF signals between sensors. To obtain accurate TDOA, the distance between sensors must be large enough (at least 1 × 2 m in a rectangular UHF sensor array of four elements), and the sample rate of acquisition and record equipment must also be high enough (generally more than 10 GHz). Therefore, the cost is quite high.

In substations, the status inspection for power equipment is one of the key contents of substation operation and maintenance. An intelligent inspection robot [11,12,13] is an effective way to replace manual inspection with a wide application. Existing inspection robots generally use infrared and high-definition cameras to perform temperature and status detection. However, limited by the implementation conditions of the small and lightweight antenna array, current inspection robots can hardly carry an omni-directional UHF sensor array to detect and locate the PD source. The main reason is that accurate TDOA needs a big antenna array to obtain high time delay resolution. Therefore, it is significant to design and develop a lightweight, small-sized, high-gain antenna array which can be mounted on inspection robots.

In PD source location, direction finding is relatively the most important for PD initial location. Due to this, the direction of arrival (DOA) estimation is chosen in this paper. DOA estimation methods mainly include the classical spectrum estimation method [14], the Capon minimum variance method [15], the Multiple Signal Classification (MUSIC) method and the estimating signal parameter via rotational invariance techniques (ESPRIT) method [16]. The MUSIC method is chosen in this paper for its advantage of super-resolution and exceeding the Rayleigh limit, providing a theory for the study of partial discharge positioning algorithms. In the 1970s, Doctor Schmidt from Stanford University proposed the MUSIC algorithm [17,18,19], which estimates directions based on the orthogonality of signal feature subspace and noise feature subspace. Afterward, based on the MUSIC algorithm, a large number of improved algorithms were derived [20,21,22,23,24,25], and there were also a few research studies in the field of PD detection involving the MUSIC algorithm [26,27]. However, the existing algorithms have the following limitations: (1) The essence of the MUSIC algorithm is to perform correlation calculation using the phase difference caused by the wave path difference of an array signal. If the collected signal is a UHF signal, the requirement for the sampling rate of the devices is very high. (2) The phase difference is mainly related to the signal frequency, so the algorithms are only suitable for narrow-band signals. Moreover, PD signals are ultra wide-band signals; therefore, using the MUSIC method directly will cause serious direction error. (3) The element spacing of the array cannot be greater than half of the wavelength of the incident wave; otherwise, angles will be blurred.

Based on the superiority of the spatial spectrum algorithm and the possibility of the algorithm’s migration from phase information to signal strength information, we propose the directional MUSIC (Dir-MUSIC) algorithm for DOA estimation of PD based on signal strength. In detail, we carry out the research work of signal model derivation, Dir-MUSIC algorithm performance, PD sensor array development and effectiveness verification. Compared to the phase information acquisition, the signal strength information acquisition can reduce the sample rate requirement and computational complexity. The contributions of the Dir-MUSIC algorithm are: (1) Signal strength information is applied for DOA estimation of PD source using the MUSIC method instead of phase information, and this change can reduce the requirement of sample rate and equipment cost under a high location accuracy. (2) The Dir-MUSIC algorithm provides an effective way for PD location applied in inspection robots. (3) A lightweight, small-sized, high-gain directional antenna array is designed and developed which can be mounted in inspection robots.

The rest of the paper proceeds as follows: Section 2 introduces an analytical expression of the antenna pattern and derives the Dir-MUSIC algorithm using an antenna gain array manifold based on signal strength information. In Section 3, simulation results under the condition of SNR and array manifold error are provided. Section 4 presents the PD localization experimental results with the designed directional spiral antenna circular array, and the conclusions are outlined in Section 5.

## 2. Dir-MUSIC Algorithm

The traditional MUSIC algorithm is only suitable for narrow-band signals. The reason is that the array manifold depends on signal frequency. If directional antennas are used instead of omni-directional antennas and the strength array manifold is formed based on the array pattern, the limitation of narrow-band will be eliminated, which will significantly expand the range of application of the MUSIC algorithm. Suppose the antenna array is a uniform circular array of N elements (directional antenna), and the noise is Gaussian white noise with a mean value of zero, which is independent of the signal. Also, suppose the array is in the far-field area of the source radiation; that is, what the antenna array receives from the source is a plane wave.

### 2.1. Derivation

Any continuous function can be approximated by a linear combination of Gaussian functions. Based on the actual antenna pattern, a linear combination of thre Gaussian functions is enough to describe the two-dimensional antenna pattern function, as shown in (Equation 1):(1)$$g\left(\theta \right)=a_{1}e^{-\frac{{\left(\theta -b_{1}\right)}^{2}}{c_{1}^{2}}}+a_{2}e^{-\frac{{\left(\theta -b_{2}\right)}^{2}}{c_{2}^{2}}}+a_{3}e^{-\frac{{\left(\theta -b_{3}\right)}^{2}}{c_{3}^{2}}},$$
where $a_{1}=0.5255,\;b_{1}=218.1,\;c_{1}=51.73,\;a_{2}=0.3405,\;b_{2}=304.8,\;c_{2}=41,\;a_{3}=0.6251,\;b_{3}=156.1,\;c_{3}=109.1$. These parameters are calculated according to the measured antenna pattern. The normalized antenna gain array manifold(AGAR) with six elements is shown in Figure 1.

Suppose the source signal is s(t), which is a broadband signal, and the direction of the incoming wave is θ, then the gain matrix of each element for the incoming wave in the direction is shown in (Equation 2):(2)$$G={\left[g_{1}\left(\theta \right),g_{2}\left(\theta \right),\cdots ,g_{N}\left(\theta \right)\right]}^{T},$$
where the response of each element is supposed to be approximately equal in the same direction and at different frequencies in the frequency band of interest.

The response of each element to the source is shown in (Equation 3):(3)$$X=GS={\left[g_{1}\left(\theta \right)s\left(t\right),g_{2}\left(\theta \right)s\left(t\right),\cdots ,g_{N}\left(\theta \right)s\left(t\right)\right]}^{T}.$$

Considering the noise effect, the matrix form is shown in (Equation 4)
(4)X=GS+N,
where S=s(t), $N={\left[n_{1}\left(t\right),n_{2}\left(t\right),\cdots ,n_{N}\left(t\right)\right]}^{T}$ and $n_{i}\left(t\right)$ is independent identically Gaussian distributed.

Perform covariance processing on the array output *X* to obtain the covariance matrix $R_{X}$:(5)$$\begin{array}{c} \begin{array}{cc} R_{X} & =E(XX^{T})\\ \; & =E(\left(GS+N\right){\left(GS+N\right)}^{T})\\ \; & =GE\left(SS^{T}\right)G^{T}+0+0+E\left(NN^{T}\right)\\ \; & =GR_{S}G^{T}+R_{N}, \end{array} \end{array}$$
where $R_{S}=E\left(SS^{T}\right)$ is the correlation matrix of the signal, and $R_{N}=E\left(NN^{T}\right)=\sigma^{2}I$ is the correlation matrix of the noise.

Suppose the direction angle of the incoming wave relative to element 1 is $\theta_{1}$, then the array gain matrix G of a uniform circular array of *N* elements can be obtained, that is, the strength array manifold is shown in (Equation 6):(6)$$G=\left[\begin{array}{c} g_{1}\left(\theta \right)\\ g_{2}\left(\theta \right)\\ \vdots \\ g_{N}\left(\theta \right) \end{array}\right]=\left[\begin{array}{c} a_{1}e^{-\frac{{\left(\theta_{1}-b_{1}\right)}^{2}}{c_{1}^{2}}}+a_{2}e^{-\frac{{\left(\theta_{1}-b_{2}\right)}^{2}}{c_{2}^{2}}}+a_{3}e^{-\frac{{\left(\theta_{1}-b_{3}\right)}^{2}}{c_{3}^{2}}}\\ a_{1}e^{-\frac{{\left(\theta_{1}+\frac{360}{N}-b_{1}\right)}^{2}}{c_{1}^{2}}}+a_{2}e^{-\frac{{\left(\theta_{1}+\frac{360}{N}-b_{2}\right)}^{2}}{c_{2}^{2}}}+a_{3}e^{-\frac{{\left(\theta_{1}+\frac{360}{N}-b_{3}\right)}^{2}}{c_{3}^{2}}}\\ \vdots \\ a_{1}e^{-\frac{{\left(\theta_{1}+\frac{360}{N}\left(N-1\right)-b_{1}\right)}^{2}}{c_{1}^{2}}}+a_{2}e^{-\frac{{\left(\theta_{1}+\frac{360}{N}\left(N-1\right)-b_{2}\right)}^{2}}{c_{2}^{2}}}+a_{3}e^{-\frac{{\left(\theta_{1}+\frac{360}{N}\left(N-1\right)-b_{3}\right)}^{2}}{c_{3}^{2}}} \end{array}\right]$$

For a single PD source, if the direction of the incoming wave is determined, it is impossible for each element of *G* to be 0. Therefore, the strength array manifold matrix is full rank. Obviously, $R_{S}$ is the square of modulus of the acquired signal, i.e., the signal power, whose value is greater than 0. $R_{N}$ is the correlation matrix of noise, and thus it is full-rank. Then it is obvious that $R_{X}$ is a full-rank matrix, and $rank(R_{x})=N$. In addition,
(7)$$R_{X}=E\left(XX^{T}\right)=E\left[{\left(XX^{T}\right)}^{T}\right]=R_{X}^{T},$$
thus $R_{X}$ is a Hermite matrix. According to the characteristics of a Hermite matrix, eigenvalues of the Hermite matrix are all real numbers. Because $R_{S}$ is positive definite, $R_{X}$ is also a positive definite matrix. Suppose the eigenvalues of $R_{X}$ are $\lambda_{1},\lambda_{2},\cdots ,\lambda_{N}$, and the corresponding eigenvectors are $\nu_{1},\nu_{2},\cdots ,\nu_{N}$. Since $R_{X}$ is a Hermite matrix, the eigenvectors are orthogonal to each other, that is:(8)$$\nu_{i}\nu_{j}^{T}=0,i\neq j.$$

Because there is only one signal source, there is one eigenvalue related to the signal, which is equal to the sum of the eigenvalue of $GR_{s}G^{T}$ and $\sigma^{2}$. The remaining *N* − 1 eigenvalues are related to the noise, and their values should be $\sigma^{2}$ in theory. Thus, the *N* − 1 eigenvalues related to the noise are relatively small. Using this characteristic, sort the eigenvalues of $R_{X}$ from largest to smallest:(9)$$\lambda_{1}\geq \lambda_{2}\geq \cdots \geq \lambda_{N}.$$

According to the eigenvalue definition, it has
(10)$$R_{X}\nu_{i}=\lambda_{i}\nu_{i},i=1,\cdots ,N.$$

Let $\lambda_{j}=\sigma^{2}$ is a smaller eigenvalue related to the noise, it also has
(11)$$R_{X}\nu_{j}=\lambda_{j}\nu_{j},i=2,\cdots ,N,$$
that is
(12)$$\left(GR_{S}G^{T}+\sigma^{2}I\right)\nu_{j}-\sigma^{2}I\nu_{j}=0.$$

It can be obtained by some simplification that:(13)$$G^{T}\nu_{j}=0,i=2,\cdots ,N.$$

Equation (Equation 13) suggests that the column vector of matrix G and the eigenvector corresponding to the noise space are orthogonal. Because matrix *G* contains the information of the incoming wave, (Equation 13) can be used as the basis to find the direction of the incoming wave. Form the N−1 eigenvectors related to the noise to construct the noise matrix:(14)$$E_{n}=\left[\nu_{2},\cdots ,\nu_{N}\right].$$

Let the space spectrum be (Equation 15):(15)$$P_{mu}\left(\theta \right)=\frac{1}{g^{T}\left(\theta \right)E_{n}E_{n}^{T}g\left(\theta \right)}=\frac{1}{∥E_{n}^{T}{g\left(\theta \right)∥}^{2}}.$$

When the maximum value of $P_{m}u\left(\theta \right)$ is taken, it indicates that g(θ) is orthogonal to the noise matrix. That is to say, θ is the estimated direction angle of the incoming wave.

### 2.2. CRLB of the Direction Estimation

Next, we will calculate the Cramer–Rao Low Bound (CRLB) of the direction estimation in the proposed Dir-MUSIC algorithm. Equation (Equation 4) can be written as another form:(16)y=ax+n,
where $y={\left[y_{1},y_{2},\cdots ,y_{N}\right]}^{T}$ is a single snapshot data, related to the direction θ and the noise power $\sigma^{2}$, $a=a\left(\theta \right)=[a_{1}\left(\theta \right),a_{2}\left(\theta \right),\cdots ,a_{N}{\left(\theta \right)}^{T}]$, $n={\left[n_{1},n_{2},\cdots ,n_{N}\right]}^{T}$, and *x* is the signal. Then the likelihood function and the logarithmic likelihood function can be written as:(17)$$L=f\left(y;a,\theta \right)=\frac{1}{\sqrt{2\pi }\sigma }e^{-\frac{1}{2\sigma^{2}}{\left(y-ax\right)}^{T}\left(y-ax\right)},$$
(18)$$\ln L=-\ln\left(\sqrt{2\pi }\sigma \right)-\frac{1}{2\sigma^{2}}\left(y^{T}y-y^{T}ax-x^{T}a^{T}y+x^{T}a^{T}ax\right).$$

The Fisher matrix *F* can be calculated as:(19)$$F=\left[\begin{array}{cc} -E(\frac{\partial^{2}\ln L}{\partial \sigma^{2}}) & -E(\frac{\partial^{2}\ln L}{\partial \sigma \partial \theta })\\ -E(\frac{\partial^{2}\ln L}{\partial \theta \partial \sigma }) & -E(\frac{\partial^{2}\ln L}{\partial \theta^{2}}) \end{array}\right].$$

Through calculation, we can obtain
(20)$$\left\{\begin{array}{c} E\left(\frac{\partial^{2}\ln L}{\partial \sigma^{2}}\right)=\frac{1}{\sigma^{2}}\\ E\left(\frac{\partial^{2}\ln L}{\partial \sigma \partial \theta }\right)=E\left(\frac{\partial^{2}\ln L}{\partial \theta \partial \sigma }\right)=0\\ E\left(\frac{\partial^{2}\ln L}{\partial \theta^{2}}\right)=-\frac{1}{\sigma^{2}}∥\frac{\partial a}{\partial \theta }{∥}^{2}{∥x∥}^{2}. \end{array}\right.$$

Generally, $\frac{{∥x∥}^{2}}{\sigma^{2}}$ can represent signal to noise ratio (SNR), thus CRLB is
(21)$$var\left(\theta \right)\geq \frac{1}{∥\frac{\partial a}{\partial \theta }{∥}^{2}SNR}$$

Thus, when directivity is sharper and SNR is higher, the direction finding is more accurate.

## 3. Simulation Research on Algorithm Performance

### 3.1. Relation between Incoming Wave Direction Estimation and SNR

To explore the performance of the algorithm under different SNR, the Dir-MUSIC algorithm is simulated and tested on the MATLAB platform to verify its reliability. The simulation process is shown in Figure 2.

The simulation settings are shown as follows. First, the strength array manifold is constructed based on the array element pattern, as shown in Figure 1. Then, 3600 incoming wave directions are randomly generated in the range of the circle $\left[1^{{}^{\circ}},360^{{}^{\circ}}\right]$, and 3600 simulations are performed. The accuracy of incoming wave direction estimation is defined as: if the error between the estimated angle and the true value is less than $2^{{}^{\circ}}$, the positioning is successful; the accuracy is equal to the ratio of the number of successful positioning to the total number of simulations. The double exponential oscillation attenuation function was used to simulate the PD pulse signal [28], which is shown in (Equation 22):(22)$$f\left(t\right)=\left(e^{-\frac{1.3t}{k_{1}}}-e^{-\frac{2.2t}{k_{2}}}\right)\sin\left(2\pi tf\right),$$
where $k_{1}=1.5\times 10^{-8},k_{2}=2.0\times 10^{-8},f=1.0\times 10^{8}$, and the received signals of the array after adding noise are shown in Figure 3.

The accuracy of the incoming wave direction estimation for different SNR is shown in Table 1.

It can be seen from Table 1 that the proposed algorithm has a strong anti-interference effect on Gaussian white noise. When the SNR drops from 10 to −5, the estimation accuracy of the DOA could still reaches 99.17%. When the SNR was −10, as shown in Figure 3, the PD pulses of channel 5 and channel 6 had been completely covered by noise, and the PD pulses of channel 1 and channel 4 were not obvious. At this time, the accuracy rate was still up to 72.78%. The figure below shows the angle estimation error for different incoming wave directions when the SNR was −10.

As shown in Figure 4, the upper limit of the error distribution was 6, the lower limit was −6, the mean was −0.1139, and the variance was 2.3404. The distribution of angle estimation errors was relatively concentrated, which met the positioning requirements.

### 3.2. Research on Direction-Finding Performance under the Condition of Array Manifold Error

When the antenna array works under the ideal model, the Dir-MUSIC algorithm surely has excellent direction-finding performance. However, when errors exist in the array manifold, the performance of the algorithm will rapidly decrease or even fail. In practice, due to the antenna manufacturing process, signal acquisition channels, working environment and other factors, the amplitude error (mainly between the actual *G* and the estimated *G*) greatly affects the direction-finding performance of the proposed algorithm. When the array has an amplitude error, the array manifold for six antennas becomes
(23)$$G^{\prime }={\left[g_{1}\left(\theta \right)+\Delta g_{1},\cdots ,g_{6}\left(\theta \right)+\Delta g_{6}\right]}^{T}.$$

Then the actual received data is:(24)$$X=G^{\prime }S+N.$$

In the subsequent spectrum search process, the estimated *G* is still used as the array manifold from calculation, which caused errors in direction-finding. This section will explore the impact of different amplitude errors on direction-finding performance. In this section, uniform distributed random errors are added to the array manifold. Respectively, U(−0.1,0.1),U(−0.075,0.075),U(−0.05,0.05) and U(−0.025,0.025) were added to form the array manifold $G^{\prime }$ with errors, and the received signal *X* was generated using $G^{\prime }$, but the direction finding was performed based on the array manifold *G* without errors. In the case of SNR of 10, 3600 Monte Carlo simulations were performed, and the simulation results are shown in Figure 5.

It can be seen from Figure 5 that when the error increased from U(−0.025,0.025) to U(−0.1,0.1), the direction-finding accuracy decreased from 98.30% to 42.89%, and the standard deviation of the error increased from $1.13^{{}^{\circ}}$ to $4.29^{{}^{\circ}}$. However, the standard deviation of the error was about θ, which did not change with the amplitude error. The mean value was almost unchanged because added error obeys uniform distribution. The estimated error of θ in (Equation 15) was nearly proportional to the added error. The greater the added error, the greater the estimated error of θ was. Therefore, the mean value of the direction-finding error did not change with the value of the added error, and the standard deviation of the direction-finding error increased with the increase of the added error. When the added error was U(−0.1,0.1), the standard deviation of the estimated error was only $4.29^{{}^{\circ}}$, which showed that the proposed algorithm had good robustness to the amplitude error of the array.

### 3.3. Relation between Incoming Wave Direction Estimation and Number of Array Element

Direction-finding performance is also strongly related to the number of array elements. In traditional spatial spectrum estimation, the number of array elements determines the array aperture. The larger the number of array elements, the larger the array aperture is, and the direction-finding performance is better. On the premise that the antenna pattern of a single element is determined, the number of elements determines the array manifold. Due to the miniaturization requirement of the array, the radius of the array cannot exceed the radius R of the inspection robot chassis, which is about 20 cm. Assuming that the radius *R* of the uniform circular array is constant, then the number of array elements affects the distance between adjacent array elements, and different array element distances make different steering vectors, as shown in Figure 6.

In Figure 6a, the four-element array (blue) has an element pitch angle of $90^{{}^{\circ}}$, and the eight-element array (black) has an element pitch angle of $45^{{}^{\circ}}$. The corresponding array manifolds are shown in Figure 6b,c, respectively. The increasing number of array elements results in greater data redundancy of the signal receiving matrix and stronger gain performance of the array. Theoretically, direction-finding performance seems better with a larger number of elements. However, due to the large size of the UHF antenna, if the array elements are too close, the mutual coupling among the array elements will affect the antenna performance. Therefore, it is necessary to select the optimal number of array elements to ensure direction-finding performance while considering both the array size and the single element performance.

Set the number of array elements to be 1, 2, 4, 6, 8 and 10, respectively, the channel amplitude error to be U(−0.05,0.05), the signal-to-noise ratio to be 10, and the direction of the incoming wave was randomly generated within a $360^{{}^{\circ}}$ circle. A total of 3600 Monte Carlo simulations were carried out, and the results are as shown in Figure 7.

The results suggest that the accuracy of direction-finding is very sensitive to the number of elements. When the number of array elements was increased from 1 to 10, the direction-finding accuracy continuously increased from 1% to 86.06%. This is because when the number of array elements was too small, taking a single-element array as an example, the number of elements was equal to the number of incident waves, so the covariance matrix of the received data could not be decomposed into signal space and noise space, and the orthogonality of the two spaces could not be used for direction finding, resulting in the failure of the MUSIC algorithm. With the increase of the number of array elements, the redundancy of the received data increased. The covariance matrix of the data could be decomposed into signal space formed by the eigenvectors corresponding to one signal and noise space formed by the eigenvectors corresponding to the noise. Multiple linearly independent noise vectors could make the noise space more robust to finite data length and amplitude errors. At the same time, when the number of elements increased, the gain performance of the array was also significantly improved, so the estimation of the noise space using received data was more accurate, and the direction-finding accuracy increased with the increase of the number of array elements.

When too many elements were collected in the array, on the one hand, it would cause difficulties in data collection and processing. The data amount of the covariance matrix of the received data was proportional to the square of the number of elements. Too many elements would result in more computational burden during the decomposition of the covariance matrix, which affected the real time performance of the algorithm. On the other hand, due to the limitation of the maximum radius of the array, too many array elements would reduce the distance among the array elements, and increase the mutual coupling error of the array. Therefore, generally considering the direction-finding accuracy, the statistical distribution law of the direction-finding error, the ability of data acquisition processing and the array performance, the six-element uniform circular array layout was chosen as the optimal formation.

From the pattern of a single element, it can be seen that when the angle deviated greatly from the front of the antenna, the gain was attenuated rapidly. Therefore, when waves arrived in these directions, the signals sensed by the antenna were almost completely submerged in noise. Also, in practical applications, four-channel data acquisition cards or digital oscilloscopes are often used. Therefore, the algorithm performance of the four-element array was studied according to the element interval of the six-element uniform circular array. The SNR were set, respectively, 10, 5, 0, −5, −10 to explore the direction finding accuracy of the algorithm, and the results are shown in Table 2. It can be seen from Table 2 that when the SNR was high, both the four-array element and the six-array element could maintain a high direction-finding accuracy. When the SNR was reduced to −10, the direction-finding accuracy of the four-array element dropped rapidly to 47.67%. Therefore, in practical applications, in an environment with a high SNR, a four-element array can be used for direction finding.

### 3.4. Comparison with the Method Based on Phase Information

In this subsection, an accuracy comparison between the proposed method and the method based on the phase information is carried out. In [26], a DOA estimation for PD direction finding in an air-insulated substation based on phased array theory has been proposed. In the numerical simulation, we reconstructed the method in [26] for a six-elements omni-directional antenna array with uniform circular arrangement, and the sample rate was set to 2.5G Hz. The direction finding accuracy under different SNR is shown in Table 3.

Obviously, the accuracy of the proposed method is the same as in Table 1. Under low SNR, the proposed method is better than the method in [26]. We must recognize that this comparison is a little unfair because the conditions of the two methods are very different. However, this comparison can partially verify the effectiveness of the proposed method. Additionally, the accuracy comparison between the two methods is shown in Table 4 with the same sample rate, respectively, 1 G, 500 M and 100 M. The SNR is both set to 0.

From Table 4, to obtain high direction-finding accuracy, the sample rate requirement of the phase method is much higher than the proposed method. Generally, a high sample rate requirement always means high equipment cost. The relatively low sample rate requirement is the main reason we can reduce the equipment cost. Moreover, a demodulation technique can be applied in the proposed method through hardware systems to obtain the amplitude of PD signals, which is also an effective approach to reduce the sample cost in future.

The omni-directional antenna must keep a high gain in each direction, while the directional antenna just needs to reach a high gain in one direction. Therefore, the omni-directional antenna usually costs more than the directional antenna. It can also help to reduce the total cost.

## 4. Experimental System Construction and Positioning Test

### 4.1. Experimental System Construction

UHF sensors are used to convert electromagnetic signals excited by PD into electrical signals. It is the first and most important step in PD detection and positioning. The sensitivity of the sensor directly affects the detection efficiency and positioning accuracy; therefore, design of a suitable sensor is essential. First, the sensor needs to have broadband characteristics because the electromagnetic pulse triggered by PD is a broadband signal. Second, based on the proposed Dir-MUSIC algorithm, the sensor needs to have good directional and high gain characteristics. Finally, the used sensor array needs to meet the miniaturization requirements. Based on the above design requirements of sensors, using CST simulation software, the sensor array was designed as shown in Figure 8. The main purpose of the CST simulation is to confirm the directionality of antennas, which is the key point of the proposed algorithm.

The pattern of the directional helical antenna array is very important to the direction-finding performance of the proposed algorithm. The microwave anechoic chamber can shield external electromagnetic interference. The absorbing material inside the anechoic chamber has a good effect on electromagnetic wave reception and can effectively suppress refraction and reflection. The detection system consisted of a shielded shell with a length of 19 m, a width of 8.5 m and a height of 8.5 m, an N5225A vector network analyzer, a reflecting surface, a turntable, a power amplifier, a low noise amplifier and a feed source. The vector network analyzer was connected to the emitting feed source through the power amplifier. The emitted electromagnetic wave was converted into a plane wave by the reflecting surface and propagated to the antenna. The antenna to be tested was connected to the vector network analyzer through the low-noise amplifier for data analysis. The measurement of the XOY surface pattern of the antenna was completed by controlling the rotation of the turntable via an industrial PC. The specific detection system is shown in Figure 9.

The array detection platform is shown in Figure 10. The electromagnetic wave was emitted from the horn antenna, and the electromagnetic wave formed a plane wave after passing through the wave reflecting surface. The antenna received the plane wave and transmitted the received signal to the vector network analyzer through the coaxial cable connected to the antenna for calculating the gain of the antenna. The XOY surface pattern of the antenna was then measured by controlling the turntable. The measured pattern was the gain relative to the broadband double-ridged horn antenna (standard antenna). The actual pattern of the antenna could be obtained by comparing measured data with the data of the standard antenna. At the frequency of 1.25 GHz, the measured antenna pattern is shown in Figure 11.

The result suggests that when the frequency was 1.25 GHz, the measured pattern and the simulated pattern had a similar increase and decrease trend. The gain was the largest directly in front of the helical antenna. With the angle shift, the gain gradually decreased, showing a high gain in front of the antenna and low gain on both sides. The core element of the Dir-MUSIC algorithm based on strength information is that as the direction of the incoming wave changes, the gain of the antenna needs to change drastically, so the designed antenna can meet the requirements of the positioning algorithm.

### 4.2. Positioning Experiment

To verify the reliability of the Dir-MUSIC algorithm based on strength information and the developed antenna, an experimental platform was built as shown in Figure 12. Based on the conclusion of the previous section, in actual application, if limited by the complexity of the sampling equipment and data processing, in an environment with high SNR, using a four-element array for direction finding could reach the equivalent direction-finding accuracy of a six-element array. The experimental platform consisted of a digital oscilloscope with storage function, four equal-length radio frequency coaxial cables, the developed uniform circular array directional spiral antenna array and a discharge gun. The location of the experimental site was calibrated by measuring tools such as infrared range-finders, and PDs were simulated by the discharge gun, performed at different positions of the array. The antenna array received electromagnetic waves and stored them in the digital oscilloscope. Channels 1 to 4 corresponded to the data received by array element 1 to 4, respectively. Since the volume of the antenna was centimeter-level, its volume was negligible compared to the partial discharge source of 7.5 m and tens of meters away, so the positions of element 1 to element 4 were (0 m, $90^{{}^{\circ}}$), (0 m, $30^{{}^{\circ}}$), (0 m, $-30^{{}^{\circ}}$), (0 m, $-90^{{}^{\circ}}$); 20 sets of simulated discharges were performed at the same position.

Discharge was performed at different positions in the experimental area, with 20 times at each position. The experimental platform is shown in Figure 12. The width of the discharge pulse waveform was about 2.5 ns. Frequency domain analysis on multiple sets of received data was also performed. The energy of the pulse waveform was mainly concentrated between 1.5 GHz and 2 GHz. There was a maximum value at the frequency of 948 MHz in the spectrogram, which was preliminarily presumed to be narrow-band interference.

The positioning algorithm flowchart is shown in Figure 13. First, the array manifold is composed of the measured pattern of a single element. Then, based on digital filtering technology, the original data are subjected to band-pass, and the pulse waveform area data are intercepted and input into the algorithm module to estimate the direction of the incoming wave. In the algorithm module, the filtered data needs to be mapped to the interval [−1,1] to suppress the dependence of the received signal amplitude on the distance of the discharge source, so the received data and the array manifold have the same influence on the direction-finding results.

Using the Dir-MUSIC algorithm based on strength information to find the direction of 10 sets of experimental data, the results are shown in Table 5. It can be seen from the results that the proposed algorithm had good direction-finding performance in multiple incoming wave directions. The maximum mean value of the angle error was $7.55^{{}^{\circ}}$, the minimum was only $0.7^{{}^{\circ}}$, and the average multi-direction-finding error was $3.39^{{}^{\circ}}$. The standard deviation of the angle error was as low as $0.31^{{}^{\circ}}$, and the maximum was $7.5^{{}^{\circ}}$. Over 90% of the standard deviation of the error was less than $2.5^{{}^{\circ}}$, which proved the stability of the algorithm.

## 5. Conclusions

This paper focuses on the algorithm research based on signal strength information and carries out the research work of model derivation, performance research, sensor array development, and algorithm experimental verification of the Dir-MUSIC algorithm. The main results and conclusions of this paper are as follows:

We theoretically derived the Dir-MUSIC algorithm based on an antenna gain array manifold (AGAM). We simulated and explored the performance of the algorithm. Through simulation research, we found that under high SNR, the direction-finding accuracy of the algorithm can be as high as 100%. When the SNR drops to −10, the direction-finding accuracy can still reach 72.78%. Moreover, the AGAM error of the algorithm has good robustness.

We determined the optimal array form, comprehensively considering the direction-finding accuracy, the statistical distribution law of the direction-finding error, the ability of data acquisition processing and array performance. The six-element uniform circular array layout is the optimal formation, and the direction-finding accuracy of using a four-element array and a six-element array is equivalent when the SNR is high.

We carried out the preliminary experimental verification of the sensor system design and the Dir-MUSIC algorithm. Based on the positioning requirements, the directional antenna array was researched. The results suggest that the developed directional spiral antenna array pattern is sensitive to angle changes, and its performance meets the requirements of the positioning algorithm. Results of positioning experiments show that the proposed algorithm had good direction-finding performance in multiple incoming wave directions. The average multi-direction-finding error is $3.39^{{}^{\circ}}$. Moreover, the dispersion of direction-finding results was small, which meets the positioning requirements.

However, there still exist some limitations in the Dir-MUSIC algorithm. In fact, the antenna response at different frequencies is not equal, which is ignored in this paper. In the future, we can develop a wide-band Dir-MUSIC algorithm to raise accuracy. Moreover, it must be proven whether the DOA technique and the MUSIC algorithm in this paper are optimal. In the implementation, although the directional antennas are designed with the same model, they may have a few differences in production. Therefore, the actual antenna gain of every directional antenna must be measured to ensure an accurate steering vector. 

## Figures and Tables

**Figure 1 sensors-22-05406-f001:**
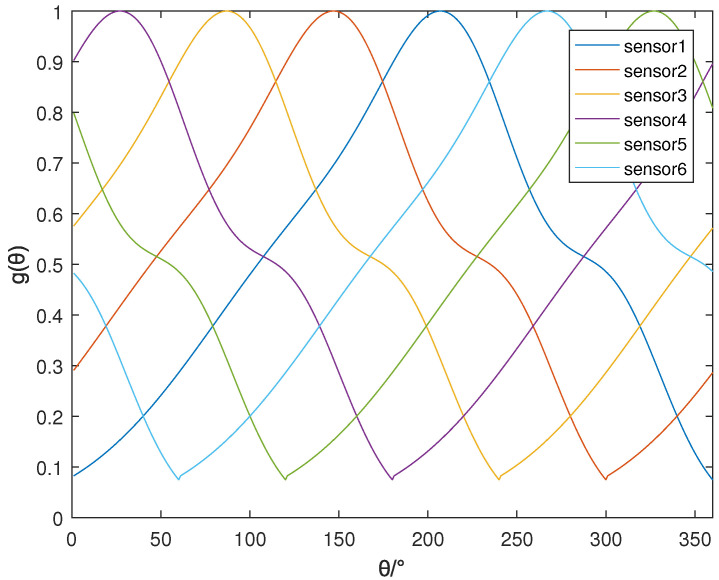
Normalized AGAR.

**Figure 2 sensors-22-05406-f002:**
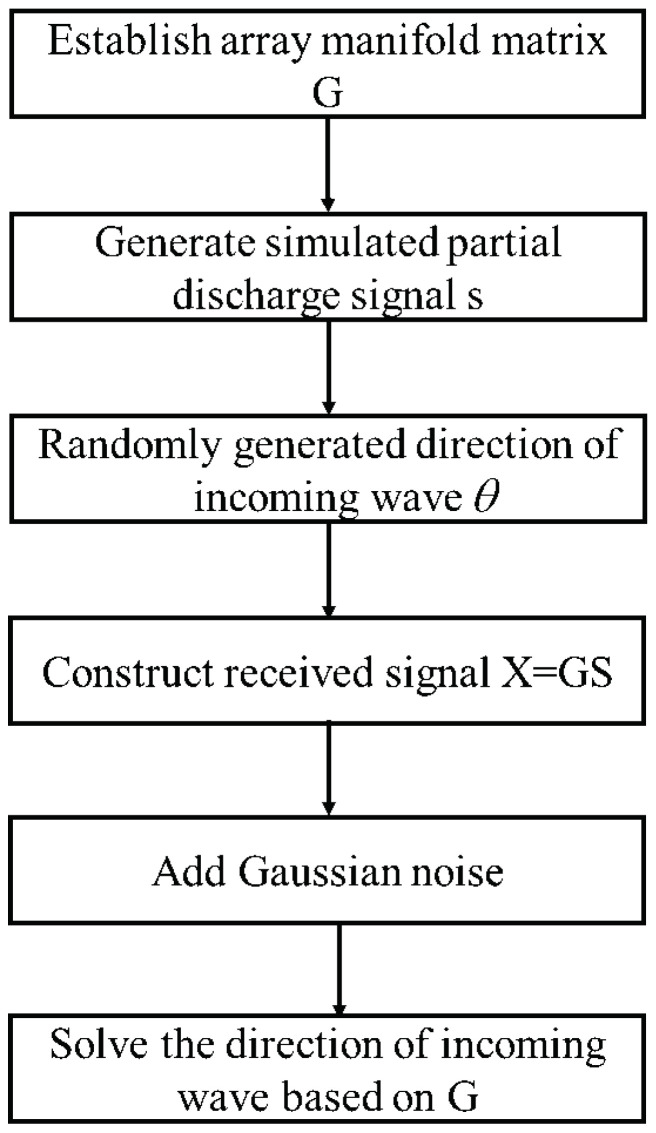
Simulation flowchart of Dir-MUSIC.

**Figure 3 sensors-22-05406-f003:**
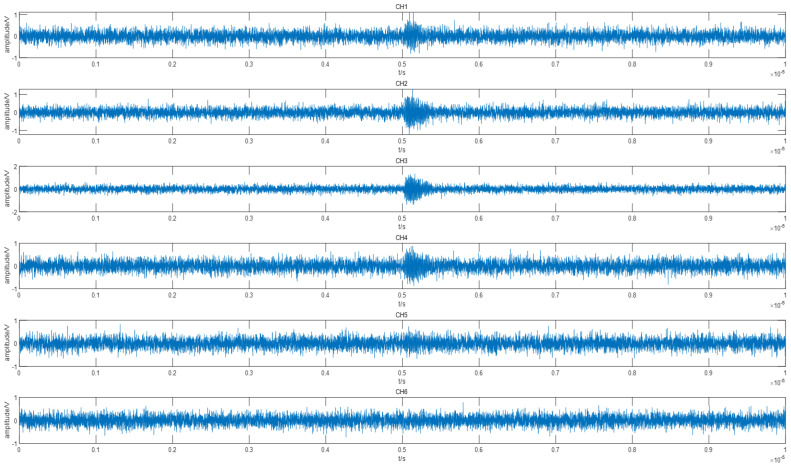
Waveform of 6-channel PD pulse after noise addition (SNR = −10).

**Figure 4 sensors-22-05406-f004:**
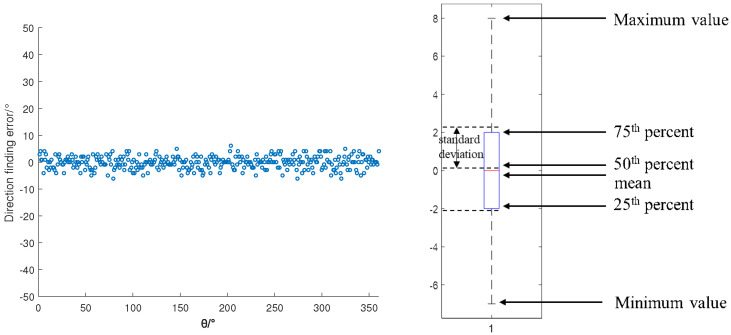
Direction-finding error in different directions (SNR = −10). (**a**) Error scatter plot. (**b**) Error box diagram.

**Figure 5 sensors-22-05406-f005:**
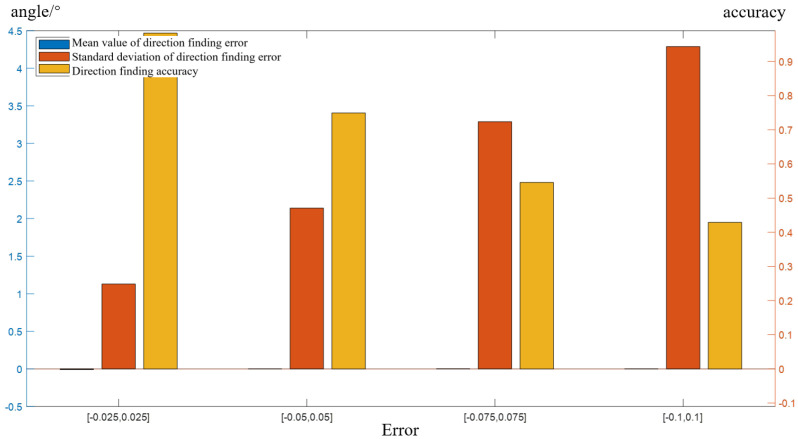
Direction-finding results under different array manifold errors.

**Figure 6 sensors-22-05406-f006:**
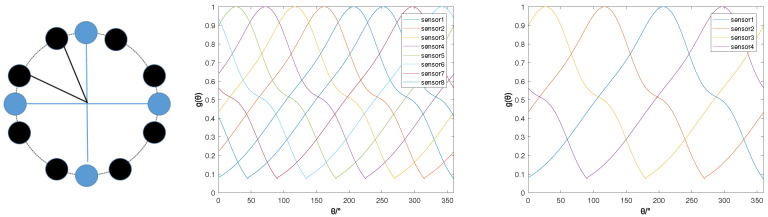
Array diagram and AGAR with different number of elements. (**a**) Array diagram. (**b**) AGAR (*N* = 8). (**c**) AGAR (*N* = 4).

**Figure 7 sensors-22-05406-f007:**
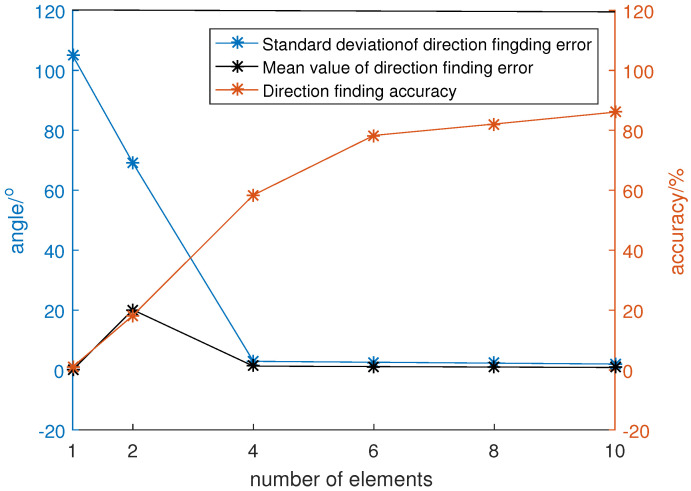
Direction-finding performance under different array elements.

**Figure 8 sensors-22-05406-f008:**
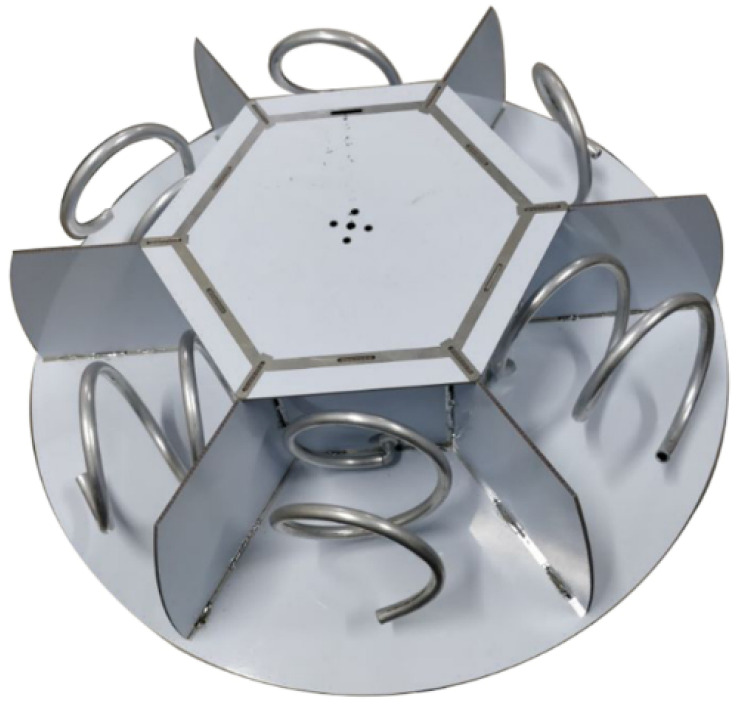
Physical picture of directional spiral antenna array.

**Figure 9 sensors-22-05406-f009:**
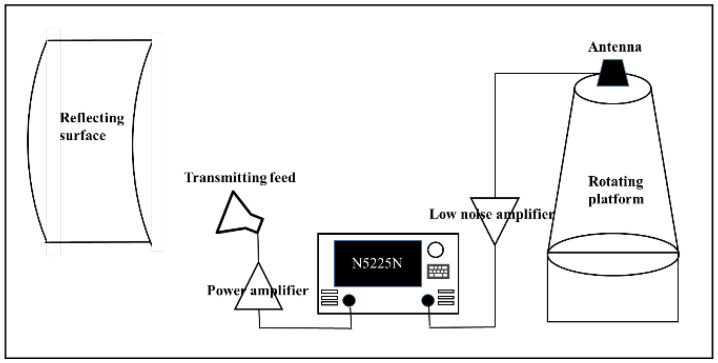
Diagram of microwave anechoic chamber detection system.

**Figure 10 sensors-22-05406-f010:**
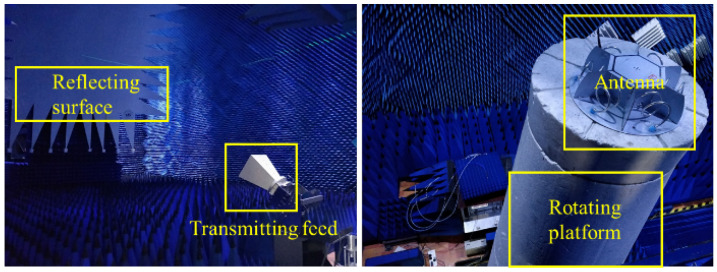
Antenna detection platform in microwave anechoic chamber.

**Figure 11 sensors-22-05406-f011:**
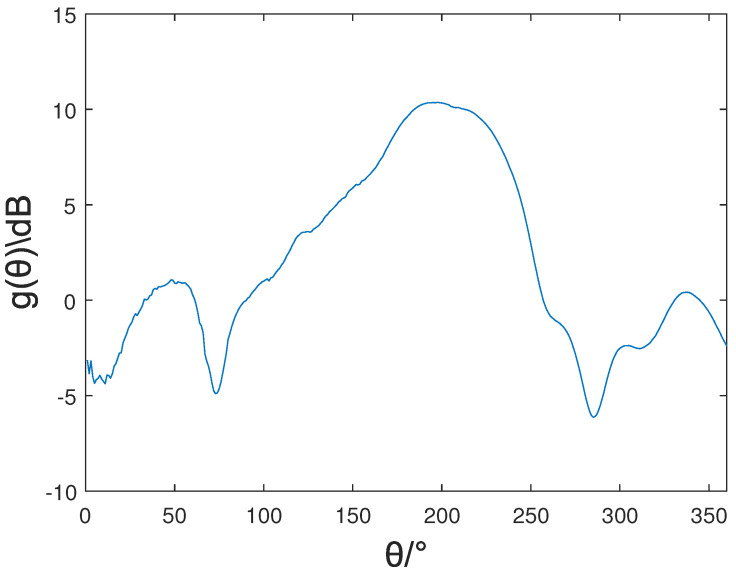
The measured pattern.

**Figure 12 sensors-22-05406-f012:**
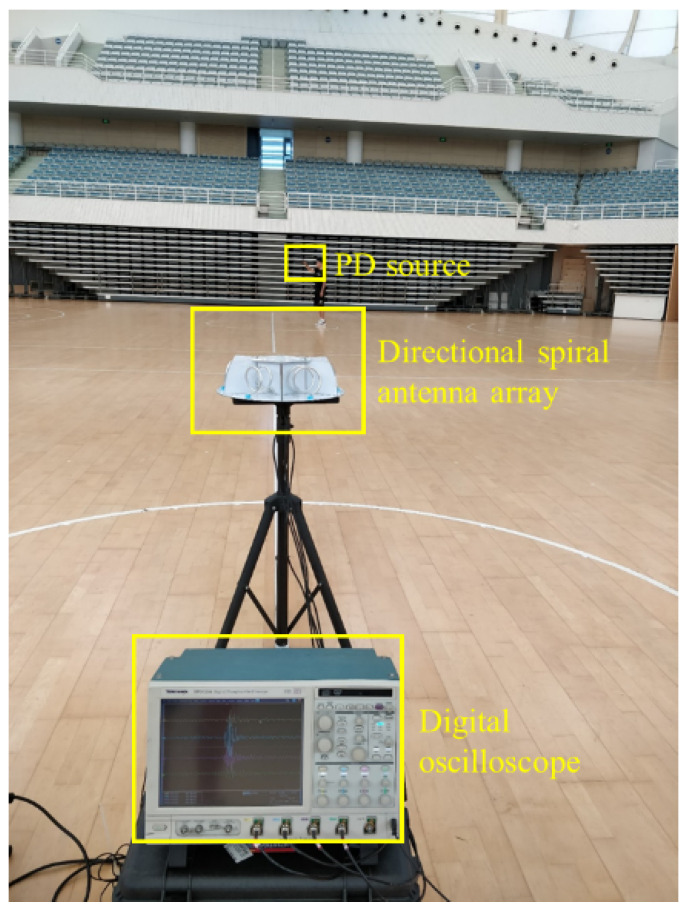
Experimental platform.

**Figure 13 sensors-22-05406-f013:**
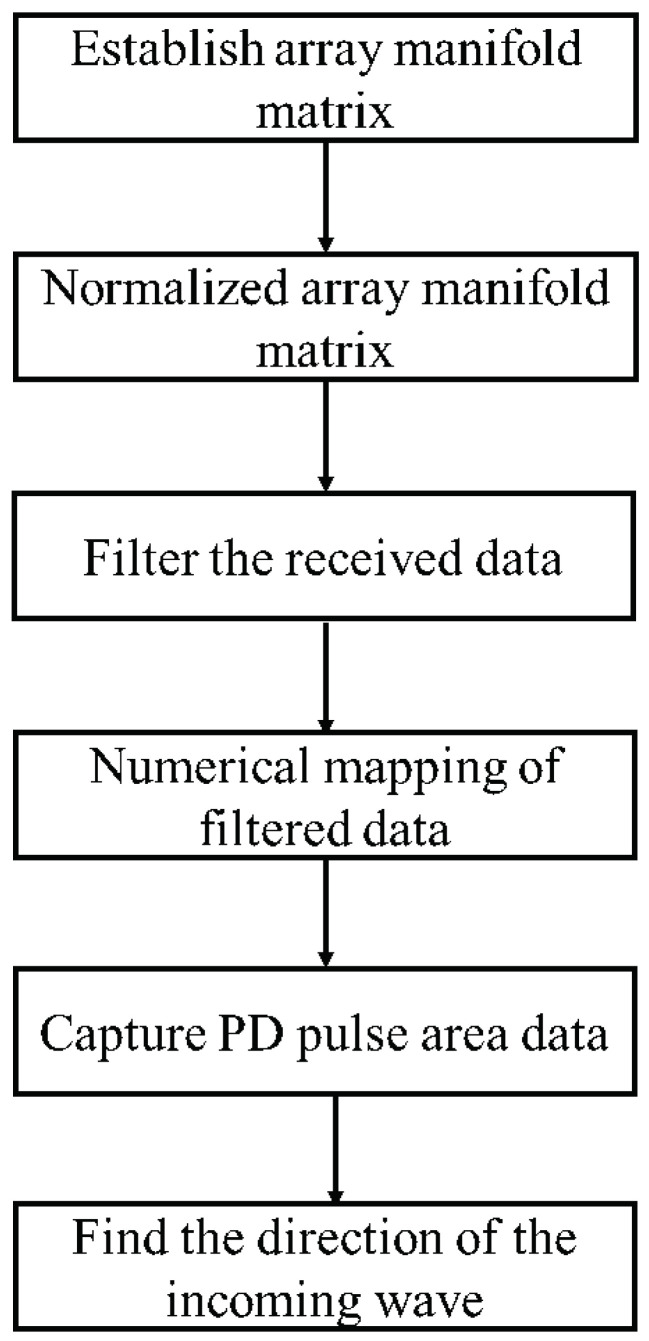
Flowchart of the Dir-MUSIC algorithm.

**Table 1 sensors-22-05406-t001:** Direction-finding accuracy under different SNR.

SNR	10	5	0	−5	−10
Accuracy	100%	100%	100%	99.17%	72.78%

**Table 2 sensors-22-05406-t002:** Direction finding accuracy under different SNR.

SNR	10	5	0	−5	−10
Accuracy	100%	100%	98.17%	85.62%	47.67%

**Table 3 sensors-22-05406-t003:** Direction finding accuracy comparation under different SNR.

SNR	10	5	0	−5	−10
Accuracy of the proposed method	100%	100%	100%	99.17%	72.78%
Accuracy of the phase method	100%	100%	97.44%	76.75%	39.00%

**Table 4 sensors-22-05406-t004:** Direction-finding accuracy comparation with sample rate changing.

Sample Rate	1 G	500 M	100 M
Accuracy of the proposed method	100%	91.64%	79.97%
Accuracy of the phase method	97.22%	72.33%	56.25%

**Table 5 sensors-22-05406-t005:** The results of the direction of 10 sets of experimental data.

Real PD Coordinates	Mean of Calculated Angle	Mean of Angle Error	Standard Deviation of Angle Error
$\left(7.5,-96^{{}^{\circ}}\right)$	$-95.25^{{}^{\circ}}$	$0.75^{{}^{\circ}}$	$1.68^{{}^{\circ}}$
$\left(7.5,-66^{{}^{\circ}}\right)$	$-73.55^{{}^{\circ}}$	$-7.55^{{}^{\circ}}$	$1.43^{{}^{\circ}}$
$\left(7.5,-19^{{}^{\circ}}\right)$	$16.8^{{}^{\circ}}$	$2.20^{{}^{\circ}}$	$0.41^{{}^{\circ}}$
$\left(7.5,-10^{{}^{\circ}}\right)$	$-17.1^{{}^{\circ}}$	$-7.10^{{}^{\circ}}$	$0.31^{{}^{\circ}}$
$\left(7.5,10^{{}^{\circ}}\right)$	$13.85^{{}^{\circ}}$	$3.85^{{}^{\circ}}$	$0.58^{{}^{\circ}}$
$\left(7.5,17^{{}^{\circ}}\right)$	$16.2^{{}^{\circ}}$	$-0.80^{{}^{\circ}}$	$0.95^{{}^{\circ}}$
$\left(7.5,28^{{}^{\circ}}\right)$	$23.65^{{}^{\circ}}$	$-4.35^{{}^{\circ}}$	$2.48^{{}^{\circ}}$
$\left(7.5,93^{{}^{\circ}}\right)$	$93.70^{{}^{\circ}}$	$0.70^{{}^{\circ}}$	$0.57^{{}^{\circ}}$
$\left(14,-90^{{}^{\circ}}\right)$	$-93.15^{{}^{\circ}}$	$-3.15^{{}^{\circ}}$	$2.11^{{}^{\circ}}$
$\left(18.9,17.5^{{}^{\circ}}\right)$	$22.05^{{}^{\circ}}$	$3.45^{{}^{\circ}}$	$7.50^{{}^{\circ}}$

## Data Availability

All data, models, or code generated or used during the study are available from the corresponding author by request.

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
