# Peer review of "Dir-MUSIC Algorithm for DOA Estimation of Partial Discharge Based on Signal Strength Represented by Antenna Gain Array Manifold"

_sensors, 2022, doi:10.3390/s22145406_

Round 1
Reviewer 1 Report
This work presents a theoretical and practical development for the DoA estimation of partial discharges. The topic (this particular application) is relevant, but it should be better motivated. Are there other alternatives than DoA estimation? If DoA is the only technique used by the community, is MUSIC the only algorithm being used?
Moreover, a more detailed discussion focused on the signal processing aspects of DoA should be included. The literature review is insufficient. I missed more relevant signal processing journal papers, as from the IEEE Trans. on Signal Processing, IEEE Signal Processing Letters, Elsevier Signal Processing. If this topic is not relevant for those journal, then perhaps this topic is then not relevant in terms of signal processing... Certainly that is not the case, it is just the literature review that is clearly insufficient.
Still on the literature review, the authors fail to make it very clear what is the novelty and contribution of this work, as the state of the art is unclear.
The proposed method is only evaluated through simulations, which is not enough. Some theoretical analysis should be included.
The numerical results should also include other alternative methods, so that the performance of the proposed scheme can be better assessed.
The limitations of the proposed method should be clearly discussed as well.
Finally, the manuscript requires a careful language review.
Reviewer 2 Report
The authors provide a method for identifying the location of partial discharge suitable for inspection robots. The background of the method, the method itself, the simulations and the physical implementation of the method is clearly described in the paper. The results are well supported and can be utilized in many fields. I only have a few questions.
A) The authors suggest an approximation of the antenna pattern using Gaussian functions (as seen in Figure 1.). However, the measured pattern is quite different from the suggested approximation (see Figure 11.). What is the cause of the difference? How does this affect the results?
B) Figure 3. The scale labels are barely visible on the figure. Other figures also suffer from low resolution.
C) Page 8, line 183: The authors state that using too many antenna elements are not beneficial due to mutual coupling between the elements. Was this coupling considered when designing the algorithm?
D) Figure 7. There are two lines with quite similar style (both are red).
E) Figure 9. The type of the VNA is provided, but there is little information on the power and low-noise amplifier. On Figure 12. the type and the bandwidth of the oscilloscope was not provided.
F) Table 3. The mean of the angle error is much higher than the standard deviation, this suggests that there is a deterministic error in the system. What can be the cause of the observed angle error?
Round 2
Reviewer 1 Report
I thank the authors for the replies and the modifications in the manuscript. However, there are still two issues that I believe should be dealt with by the authors:
- The literature review can be further improved, including more recent high quality (high impact factor) journal papers.
- The authors should compare the performance of the proposed method with some alternative methods. I understand the argument that the idea is to desing a low cost scheme, but please let us know how much performance we are losing by being "low cost". This information is very relevant.
Round 3
Reviewer 1 Report
Thanks for the new version and to the replies to my previous comments.
However, the comparison to the phase difference method is not relevant as it is. The authors should consider a setup that is adequate for the phase difference method, so that they can discuss what makes that setup more expensive or complex than the proposed low cost scheme. In the way the comparison is performed now it is excessively unfair to the phase difference scheme. Thus, the idea is to show where the cost is since you are advocating that your method achieves good performance with low cost. I am quite surprised that the authors did not understand my concern.
